# Undernutrition in children aged 0–59 months by region and over time: secondary analysis of the Burkina Faso 2012–2018 National Nutrition Surveys

Palwende Romuald Boua ![ORCID],[1,2] Toussaint Rouamba,[1] Estelle Bambara,[3] Saidou Kaboré,[3] Ella W R Compaore,[4] Boureima Ouedraogo,[5] Halidou Tinto,[1] Marie-Louise Newell,[6] Kate Ward,[6,7] Hermann Sorgho,[1] on behalf of the INPreP Study Group

PRB and TR contributed equally.

**Correspondence to**
Dr Palwende Romuald Boua; romyboua@gmail.com

## ABSTRACT

The global burden of undernutrition remains high, responsible for significant under-five mortality in resource-limited settings. Numerous sustainable development goals (SDGs) are linked to nutrition, and nationally representative nutrition surveillance is a key activity to track progress towards SDGs and guide efficient programmes.

**Objectives** The aim of this study is to look at spatial and temporal trends in undernutrition in children under 5 years age in Burkina Faso.

**Setting** We used data from annual National Nutrition Surveys using Standardised Monitoring and Assessment of Relief and Transitions methodology (anthropometry, morbidity) over 7 years (2012–2018) in Burkina Faso.

**Participants** Children of under 5 years from households selected through systemic sampling at countrywide level.

**Main outcome measures** Prevalence of stunting (height-for-age z-score, <−2), underweight (weight-for-age z-score, <−2) and wasting (weight-for-height z-score, <−2) at regional and national. We used general linear mixed models, adjusted by age, survey year, sex, presence of fever and/or diarrhoea, and poverty index to quantify the risk of undernutrition over time and by region of residence.

**Results** Between 2012 and 2018, decreases were observed overall in the prevalence of growth retardation (stunting) decreased from 33.0% (95% CI 32.3 to 33.8) in 2012 to 26.7% (95% CI 26.2 to 27.3) in 2018. Underweight reduced from 24.4% (95% CI 23.7 to 25.1) to 18.7% (95% CI 18.2 to 19.2) for the same period and wasting decreased from 10.8% (95% CI 10.3 to 11.3) in 2012 to 8.4% (95% CI 8.1 to 8.8) in 2018. However, there was substantial variation across the country, with increased risk of undernutrition in the regions of Sahel, East and Cascades primarily. High-risk regions were characterised by a lower poverty index and limited access to healthcare services.

**Conclusions** Our findings could inform national policymakers in refining and optimising resource allocation based on the identification of high-risk areas.

## BACKGROUND

Globally, undernutrition (eg, wasting as a sign of acute undernutrition, and stunting

## STRENGTHS AND LIMITATIONS OF THIS STUDY

⇒ Children morbidity is associated with all forms of undernutrition.
⇒ Poverty index is associated with chronic undernutrition but not with acute undernutrition.
⇒ Under-five boys are at higher risk of experiencing undernutrition compared with girls.
⇒ Sahel region children are at higher risk of stunting and underweight.
⇒ Our study did not cover all the National Nutrition Surveys from 2009 to 2021, due to data unavailability, therefore our finding is only relevant for the studied period. The analysis relies on survey data but not a cohort, hence causality cannot be inferred.

as a sign of chronic undernutrition) remains an important public health issue, affecting the growth and mortality of children, especially in low-income and middle-income countries.[1–3] Globally, in 2020, wasting and stunting affected an estimated 45.4 and 149.2 million children under 5 years of age, respectively.[45 6] Undernourished children have a higher risk of death from common childhood illness such as diarrhoea, pneumonia and malaria,[5 7 8] and longer-term are likely to experience poorer physical and cognitive development, increased risk of chronic disease in older age, with undernourished women at increased risk of delivering low birth weight babies.[9–11]

In Burkina Faso, like most low-income countries, improvements in nutrition through better diet quality and access to food, continues to be a major health, socioeconomic and political challenge, and is one of the main public health priorities towards achieving sustainable development goals (SDGs) 1, 2 and 3.[12] In Burkina Faso, more than one-third of deaths among children

under 5 years of age are directly or indirectly attributable to undernutrition. Despite the development and implementation by successive governments of numerous policies and programmes to deal with undernutrition, the prevalence of undernutrition (stunting, underweight and wasting) remains high. Factors such as recurrent drought, sociocultural practices (poor dietary practices), poverty, diarrhoeal and febrile diseases and insufficient access to basic health services negatively affect the nutritional status of children.[3 8 13–15] In addition, there is the problem of insecurity that the country has been experiencing since 2015, leading to massive internal displacement of the population, aggravating the already very limited access to basic livelihoods, healthcare and nutrition services in a context of poverty in these localities.[16–20] Food insecurity (lack of consistent access to enough food for every person in a household to live an active, healthy life) has been highly prevalent in the country (bulletin special FSC).

The government, supported by its financial and technical partners, has put in place a system to monitor the nutritional situation through the implementation of yearly national nutritional surveys according to the SMART (Standardised Monitoring and Assessment of Relief and Transitions) methodology. This methodology is widely used by governments and humanitarian partners to conduct timely nutrition surveys in all contexts (emergency, development, displaced populations). SMART surveys are conducted on a regular basis, often in connection with seasonal malnutrition, and can be conducted at the national or regional level, and even on a smaller scale. This paper follows a secondary analysis of the survey databases from 2012 to 2018 to have an overview of temporal trends and overall spatial risk of undernutrition and identify the areas to prioritise for resource allocation according to the risk of areas.

## METHODS
### Study setting
Burkina Faso is a landlocked country in West Africa divided into 13 regions, 45 provinces, 351 communes (49 urban and 302 rural) and 8228 villages. According to the latest census, the country has 20.5 million inhabitants, 45.3% of whom are under 15 years old and children aged 0–4 years represent 16.2% of the total population.[21] The country is ranked 182nd out of 189 according to the Human Development Index report with more than 41.4% of its population living below the poverty line.[22]

The country's epidemiological profile is marked by the persistence of a high burden of disease due to infectious disease, and by the progressive increase in the burden of non-communicable diseases. Besides infectious and non-communicable diseases, undernutrition is one of the major diseases of public health importance.[23] The food security situation is also of concern. It is estimated that approximately 3.28 million people are food insecure and in need of immediate humanitarian assistance, representing 15% of the country's total population.[24]

### Patient and public involvement
Because we used data from National Nutrition Surveys (NNSs), it was not appropriate or possible to involve patients or the public in the design, or conduct, or reporting, or dissemination plans of our research.

### Data sources
The Directorate of the Nutrition of Burkina Faso provided data (accessible through on request). The data are from annual cross-sectional NNSs using the SMART methodology. This consists of a rapid assessment survey of the food and nutrition situation among children under 5 years of age, women of childbearing age and households. For this study, we focus on anthropometric measurements collected between 2012 and 2018 (seven surveys) among the population aged 0–59 months.

We examined trends in three crude indices of nutritional problems in Burkina Faso over a 7-year period, namely stunting (height/length-for-age), underweight (weight-for-age) and wasting (weight-for-height/length). Anthropometric measurements were converted into weight-for-age z-score (WAZ), height-for-age z-score (HAZ) and weight-for-height z-score (WHZ), using the 2006 WHO child growth standards.[25] Children with oedema, which can be a sign of acute undernutrition) were excluded as recommended for this analysis (table 1).

For 2012 and 2014, the SMART survey was provincially representative for six regions (Cascades, Centre West, East, North, Sahel and South West) and regionally representative for the other seven (Boucle du Mouhoun, Centre-South, Centre, Centre-North, Centre-East, Central Plateau and Hauts-Bassins). For 2013 and 2015, the SMART survey had provincial representativeness for seven regions (Boucle du Mouhoun, Centre-South, Centre, Centre-North, Centre-East, Central Plateau and Hauts-Bassins) and regional representativeness for the other six (Cascades, Centre-West, East, North, Sahel and South-West). For 2016, the SMART survey had regional representativeness for 13 regions (resulting in smaller sample) and from 2017 onwards, the SMART survey had provincial representativeness for the 45 provinces across the 13 regions.

Each level of representativeness constituted strata.

### Outcome variables
Stunting was determined if HAZ<−2 SD, underweight as WAZ<−2 SD and wasting as WHZ<−2 SD (table 1). We excluded z-scores with extreme values from the observed mean, the following WHO indicators were used in the final analysis: WHZ −5 to 5; HAZ −6 to 6; WAZ −6 to 5.

The independent variables were individual-level characteristics mainly age in months and gender, diarrhoea and fever during the 2 weeks preceding the survey, year of the survey and cluster-level characteristics constituted by the poverty index for each province.[26]

**Table 1** Definition of anthropometric measurements and indices in children under-five

| Index | Nutritional problem measured | Indicator |
|---|---|---|
| Height/length-for-age | Severe stunting | HAZ<−3 SD |
| | Moderate stunting | HAZ<−2 SD and HAZ≥−3 SD |
| | Global stunting | HAZ<−2 SD |
| Weight-for-age | Severe underweight | WAZ<−3 SD |
| | Moderate underweight | WAZ<−2 SD and WAZ≥−3 SD |
| | Global underweight | WAZ<−2 SD |
| Weight-for-height/length | Severe wasting | WHZ<−3 SD |
| | Moderate wasting | WHZ<−2 SD and WHZ≥−3 SD |
| | Global wasting | WHZ<−2 SD |

HAZ, length/height-for-age z-score; WAZ, weight-for-age z-score; WHZ, weight-for-length/height z-score.

## Statistical analysis

For descriptive statistics at the individual and strata levels, estimates account for the complex survey design and the sampling weights provided by *survey*. Descriptive analysis of the study population was presented using means and SE for continuous variables and frequencies and proportions for categorical variables. These descriptive analyses were performed to estimate the crude temporal trend of prevalence nationwide and provide a yearly map for the regional disparities of undernutrition index.

Considering that the data have a hierarchical structure, and we would need to account for the cluster (province and region) variability as well as the individual-to-individual variability, we used general linear mixed models (GLMMs, multilevel models). GLMM (multilevel models) with binomial distribution were used to quantify the association of independent variables with each study outcome variable (stunting, underweight and wasting). We assume that the outcome variables were drawn from a binomial distribution and the residual are independent and normally distributed. The GLMM contains fixed and random effects and was formulated as follows:

$$Y_{ij} \sim \text{Bernoulli } (\pi_{ij})$$

$$\text{logit } (\pi_{ij}) = \log\left[\frac{\pi_{ij}}{1-\pi_{ij}}\right] = \beta_0 + \sum_{p=1}^{P}\beta_p x_{ij} + \sum_{q=1}^{Q}\beta_q z_j + \omega_j + \varepsilon_i$$

$$\epsilon_i \, N(0,1)$$

The random effects are also, independent and normally distributed.

In this general linear model, $Y_{ij}$ represents stunting, underweight or wasting for a child $i$ in a cluster $j$. The logit $(\pi_{ij})$ was used to model the probability of success (ie, occurrence of undernutrition) as a linear combination of observed individual characteristics $(x_{ij})$ and contextual characteristics $(z_j)$ associated with an unobserved specific effect (random effect) of the province and region $(\omega_j)$. The model contains a random error for child $i$ $(\varepsilon_i)$ that does not depend on the province/region or year. The fixed effect is composed of $\beta_0$ (intercept), $\beta_p$ (vector of regression coefficients associated with the individual-level variables) and $\beta_q$ (vector of regression coefficients associated with the cluster-level variables). The unobserved specific effect or random effect $(\omega_j)$ has two components, province specific effect and region-specific effect. These cluster specific effects were considered as province and region-specific risk.

Since we intended to quantify the association between the outcomes and dependent variables by avoiding the coefficient being affected by the individual weight, survey weights were not used to adjust the regressions and a complete case analysis was used. The model diagnostics were checked by generating a diagnostic that plots the fitted or predicted values against the residuals. All statistical analyses were performed with R statistical software (R Development Core Team, R Foundation for Statistical Computing, Vienna, Austria). The descriptive statistics that accounted for the complex survey design and sampling weights were performed using 'survey' package, regression models were fitted using the 'lme' package.

## RESULTS

### Demographic characteristics and anthropometrical measures of children

The overall number of children under-five assessed for their nutritional status using anthropometric measurement and included in the present analysis is shown in table 2. Since the survey in 2016 was set up to have regionally representative estimates, 9434 children were surveyed. Gender was equally distributed through the surveys and varied between 51.6% and 51.1% for the proportion of male children. The average age of the children during the period of 2012–2018 ranged from 27 to 29 months.

The distribution of anthropometric measures such as weight, height and the z-scores with WHO as reference population (HAZ, WAZ and WHZ) from 2012 to 2018 are presented in table 2. All z-scores had a mean value below the WHO standard. HAZ values ranged from −1.50 to −1.27; WHZ values ranged from −0.70 to −0.53; WAZ values ranged from −1.33 to −1.11.

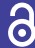

**Table 2** Characteristic of children under-five surveyed at household in Burkina Faso, 2012–2018

| Index | Weighted mean (SE) | | | | | | |
| --- | --- | --- | --- | --- | --- | --- | --- |
| | 2012 N=19 049 | 2013 N=19 170 | 2014 N=15 835 | 2015 N=20 408 | 2016* N=9434 | 2017 N=19 409 | 2018 N=23 939 |
| Age | 27.80 (0.11) | 27.15 (0.12) | 27.82 (0.13) | 27.47 (0.11) | 28.59 (0.16) | 28.53 (0.11) | 29.04 (0.10) |
| Weight | 10.46 (0.02) | 10.39 (0.02) | 10.72 (0.03) | 10.41 (0.02) | 10.85 (0.03) | 10.8 (0.02) | 10.92 (0.02) |
| Height | 82.08 (0.09) | 81.54 (0.1) | 82.68 (0.11) | 81.97 (0.09) | 83.38 (0.13) | 83.31 (0.09) | 83.75 (0.08) |
| HAZ | 1.49 (0.01) | 1.50 (0.01) | 1.30 (0.01) | 1.47 (0.01) | 1.30 (0.01) | 1.28 (0.01) | 1.27 (0.01) |
| WHZ | 0.67 (0.01) | 0.61 (0.01) | 0.53 (0.01) | 0.70 (0.01) | 0.55 (0.01) | 0.60 (0.01) | 0.56 (0.01) |
| WAZ | 1.31 (0.01) | 1.28 (0.01) | 1.11 (0.01) | 1.33 (0.01) | 1.12 (0.01) | 1.15 (0.01) | 1.11 (0.01) |

*In 2016 the sampling framework was regional and national (without provincial), hence impacting the sample size. This choice by the Ministry of Health was guided by limited resources to run the national survey.
HAZ, length/height-for-age z-score; WAZ, weight-for-age z-score; WHZ, weight-for-length/height z-score.

## Prevalence of undernutrition

The annual prevalence of chronic undernutrition (stunting) using HAZ ranged from 33.0% (95% CI 32.3 to 33.8) in 2012 to 26.7% (95% CI 26.2 to 27.3) in 2018, with a maximum of 34.2% (95% CI 33.4 to 34.9) in 2013. In other words, between 2013 and 2018, stunting prevalence declined among children under-five from 33.4% to 26.7%, a decline of 19.1% (online supplemental figure 1A).

The annual prevalence of underweight (WAZ) ranged from 24.4% (95% CI 23.7 to 25.1) in 2012 to 18.7% (95% CI 18.2 to 19.2) in 2018, with a maximum of 25.3% (95% CI 24.6 to 25.9) in 2015 (online supplemental figure 1B). This meant that nationally, underweight decreased by 23.3% between 2012 and 2018.

The annual prevalence of wasting (WHZ) (online supplemental figure 1C) ranged from 10.8% (95% CI 10.3 to 11.3) in 2012 to 8.4% (95% CI 8.1 to 8.8) in 2018, with a maximum of 11.7% (95% CI 10.7 to 11.6) in 2015. In other words, between 2013 and 2018, global wasting prevalence declined among children under-five from 33.4% to 26.7%, corresponding to a decline of 22.2%.

Overall, there was a reduction in prevalence of undernutrition over the years, but this was not seen in all areas. In other words, there was a spatial heterogeneity of undernutrition index through the country. During the 7 years, the analysis showed a permanent high prevalence (>30%) of stunting in Sahel, and East regions (online supplemental figure 2), whereas the stable high prevalence of underweight (>30%) was mainly noted in the Sahel region (online supplemental figure 3). The high prevalence of wasting was mainly noted in the Sahel and North region (online supplemental figure 4).

The predicted probabilities of stunting, underweight, wasting by child's age at survey and stratified by gender for the year 2012 to 2017 are shown in online supplemental figures 5–7. The probability of being stunted increased with age, and was higher for male than female children, but decreased over time, that is, between 2012 and 2018 (online supplemental figure 5). Similarly, predicted prevalence of underweight decreased with children's age, was greater among male children and declined over the study period, that is, from 2012 to 2018 (online supplemental figure 6). For wasting, the probability decreased with child's age and was more marked among males, but the trend over time was stable (online supplemental figure 7).

## Potential risk factors associated with stunting, underweight and wasting from 2012 to 2017 in Burkina Faso

In multivariable analysis, allowing for adjustment (table 3) males had a significantly increased risk of stunting (adjusted OR (aOR)=1.47, 95% CI 1.43 to 1.52), underweight (aOR=1.39, 95% CI 1.34 to 1.43) and wasting (aOR=1.38, 95% CI 1.31 to 1.44) than females. Compared with children aged less than 5 months, older children had higher risks of stunting and underweight, but the odds of wasting decreased significantly over the age of 24 months. Compared with the year 2012, the analysis showed that the risk of stunting, underweight and wasting in the subsequent years, that is, 2013–2018, decreased significantly overall, except for the year 2015 for wasting. Furthermore, diarrhoea was associated with high odds of stunting (aOR=1.08, 95% CI 1.05 to 1.12), underweight (aOR=1.32, 95% CI 1.27 to 1.37) and wasting (aOR=1.46, 95% CI 1.39 to 1.54). This association was similar with the fever. There was no significant association between wasting and poverty index (aOR=1.03, 95% CI 0.96 to 1.10), whereas stunting and underweight were significantly associated with the poverty index. The model diagnostics plots were presented in supplementary materials (online supplemental text and figures 8–10).

## Estimation of province and region-specific risk

The province spatial risk of stunting, underweight and wasting was shown to be heterogeneous throughout the country. Online supplemental figure 5 displays the specific risks (or odds) of each province compared with the national mean odds ratio (aOR=1).

For stunting, the provinces with high odds were located in eight provinces namely Gourma, Ioba, Kenedougou, Kossi, Lorum, Seno and Yagha provinces, whereas the

**Table 3** Potential risk factors associated with stunting, underweight and wasting from 2012 to 2017 in Burkina Faso

| Predictors | Stunting aOR (95% CI) | P value | Underweight aOR (95% CI) | P value | Wasting aOR (95% CI) | P value |
|---|---|---|---|---|---|---|
| Gender | | | | | | |
| Female | 1 | | 1 | | 1 | |
| Male | 1.39 (1.35 to 1.43) | <0.001 | 1.31 (0.27 to 1.35) | <0.001 | 1.37 (1.31 to 1.43) | <0.001 |
| Age in months | | | | | | |
| 0–5 | 1 | | 1 | | 1 | |
| 6–11 | 1.51 (1.39 to 1.63) | <0.001 | 1.96 (1.82 to 2.11) | <0.001 | 1.67 (1.53 to 1.82) | <0.001 |
| 12–23 | 3.74 (3.49 to 4.00) | <0.001 | 2.56 (2.39 to 2.73) | <0.001 | 1.53 (1.41 to 1.65) | <0.001 |
| 24–35 | 4.94 (4.62 to 5.28) | <0.001 | 2.28 (0.13 to 2.44) | <0.001 | 0.83 (0.76 to 0.90) | <0.001 |
| 36–47 | 4.57 (4.27 to 4.89) | <0.001 | 1.67 (1.56 to 1.79) | <0.001 | 0.46 (0.41 to 0.50) | <0.001 |
| 48–59 | 3.44 (3.20 to 3.68) | <0.001 | 1.58 (1.47 to 1.69) | <0.001 | 0.49 (0.44 to 0.54) | <0.001 |
| Year of survey | | | | | | |
| 2012 | 1 | | 1 | | 1 | |
| 2013 | 0.89 (0.85 to 0.94) | <0.001 | 0.79 (0.75 to 0.84) | <0.001 | 0.72 (0.66 to 0.78) | <0.001 |
| 2014 | 0.78 (0.75 to 0.82) | <0.001 | 0.72 (0.69 to 0.76) | <0.001 | 0.73 (0.68 to 0.79) | <0.001 |
| 2015 | 0.87 (0.82 to 0.91) | <0.001 | 0.95 (0.90 to 1.00) | 0.063 | 1.00 (0.92 to 1.08) | 0.902 |
| 2016 | 0.73 (0.69 to 0.78) | <0.001 | 0.74 (0.69 to 0.79) | <0.001 | 0.70 (0.63 to 0.77) | <0.001 |
| 2017 | 0.67 (0.64 to 0.71) | <0.001 | 0.71 (0.68 to 0.75) | <0.001 | 0.86 (0.80 to 0.93) | <0.001 |
| Diarrhoea | | | | | | |
| No | 1 | | 1 | | 1 | |
| Yes | 1.19 (1.14 to 1.23) | <0.001 | 1.36 (1.31 to 1.42) | <0.001 | 1.39 (1.32 to 1.47) | <0.001 |
| Fever | | | | | | |
| No | 1 | | 1 | | 1 | |
| Yes | 1.08 (1.05 to 1.12) | <0.001 | 1.31 (1.27 to 1.35) | <0.001 | 1.48 (1.41 to 1.55) | <0.001 |
| Poverty index in % | 1.17 (1.09 to 1.26) | <0.001 | 1.15 (1.08 to 1.22) | <0.001 | 1.03 (0.96 to 1.10) | 0.456 |
| Random effects | | | | | | |
| $\sigma^2$ | 3.29 | | 3.29 | | 3.29 | |
| $\tau_{00}$ | | | | | | |
| Provinces | 0.03 | | 0.03 | | 0.04 | |
| Region | 0.03 | | 0.01 | | 0.00 | |
| ICC | 0.02 | | 0.01 | | 0.01 | |
| N region | 13 | | 13 | | 13 | |
| N provinces | 45 | | 45 | | 45 | |
| Observations | 83 161 | | 83 161 | | 83 161 | |

aOR, adjusted OR.

provinces with lower odds were located in four provinces namely, Bales, Passore, Tuy and Zounweogo provinces (figure 1A).

Regarding underweight, the provinces with high odds were located in nine provinces namely Gourma, Ioba, Kossi, Kourweogo, Lorum, Oudalan, Seno, Tapoa, Yagha provinces, whereas the provinces with lower odds were located in four provinces namely, Passore, Poni, Sourou and Zounweogo provinces (figure 1B).

Regarding wasting, the provinces with high odds were located in 11 provinces namely, Bam, Ioba, Kossi, Kourweogo, Lorum, Nayala, Oudalan, Sanguie, Seno, Tapoa and Yatenga provinces whereas, the provinces with lower odds were located in eight provinces namely, Boulgou, Comoe, Houet, Koulpelgo, Leraba, Poni, Sourou and Zounweogo province (figure 1C).

Further, the analysis also indicated some regions where the odds were statistically significant higher (Sahel and Cascades) or lower (Centre-Sud, Centre-Ouest and Boucle de Bouhoun) for stunting (figure 2A). For underweight, only the region of the Sahel exhibited higher

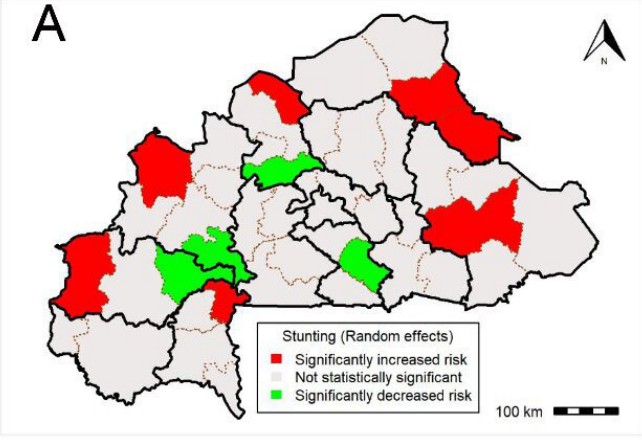

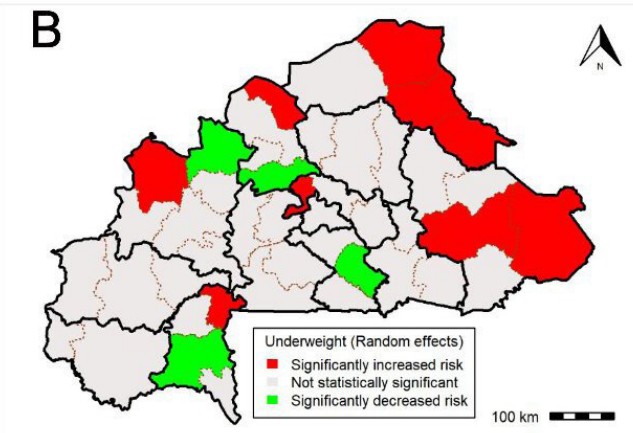

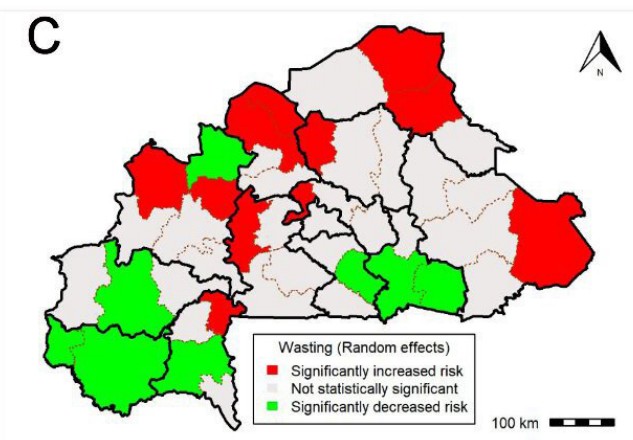

**Figure 1** Overall province-specific ORs (province random effect) of stunting (A), underweight (B) and wasting (C) among study population. Maps created by Boua *et al*.

odds (figure 2B). Finally, regarding wasting, no region exhibited a higher or lower risk (figure 2C).

## DISCUSSION

This study undertook a secondary analysis using data from annual cross-sectional NNS using SMART methodology to generate estimates that could help national policymakers in refining and optimising resource allocation based on identification of high-risk areas. The analysis found that undernutrition remains very high in certain

part of the country, although there was an overall decrease in prevalence and risk over the years in the country; the geospatial risk to experience an undernutrition event is not uniform throughout the country. This decrease in all undernutrition prevalence has been observed in other sub-Saharan Africa countries despite predicted stagnation in the region for stunting until 2020.[27–29]

In previous analysis, de Onis and collaborators predicted that in Africa, the prevalence of stunting was going to stagnate in the next coming decade from 2010, and specifically in Western Africa.[29] Our results showed a decrease in stunting prevalence of 6.7 between 2012 and 2018. de Onis prediction were based on poor data from some countries with lag time of about 10 years between them (namely Demographic and Health Survey), and the landscape of interventions in nutrition was poor. From 2009, a nutrition surveillance system has been implemented with yearly national surveys, additionally more effective nutrition programmes to improve nutritional status across the whole population have been implemented.

Sex of child (male), 2 weeks morbidity (fever, diarrhoea) were factors associated with the three forms of undernutrition; whereas poverty measured through wealth index was associated with stunting and underweight but not wasting. In their analysis of Nigeria Demographic and Health Survey, Akombi *et al* found the same factors associated to stunting.[30] Biological and social mechanism or both has been proposed to explain the differences between boys and girls.[31] Even though 2 weeks morbidity might not be causal to chronic undernutrition, it translates the effect of morbidity and or repeated morbidity on growth retardation and underweight.

Living in certain provinces and/or regions of the country confers higher risk of suffering from undernutrition. Similar findings were reported in Côte d'Ivoire, Uganda and Nigeria.[28 30 32] In Burkina Faso, these areas were the one with the highest proportion of the population living more than 10 km away from the primary health centre (first level of referral).[23] This means that access to healthcare could be associated with nutritional status. The persistent prevalence of undernutrition translates a need, not just for the community management of wasting programmes which have been shown to be effective, but for effective prevention programmes as well as structural changes allowing reduction of inequalities.[33–36]

Sahel region had the highest risk of stunting and underweight. Prior to the recent threats from terrorism, Burkina Faso had already endured a decade of a fragile food security situation due to droughts and floods, affecting food security and nutrition, especially in the northern part of the country. Restrictions on the movement of people and goods, as well as limited access to natural resources and regional or international markets, had placed a heavy burden on local food systems and people's livelihoods. The food insecurity situation in Burkina Faso is currently aggravated by a high level of insecurity. The populations of the Sahel zone are the most affected. Undernutrition is chronic and its causes are structural and complex.

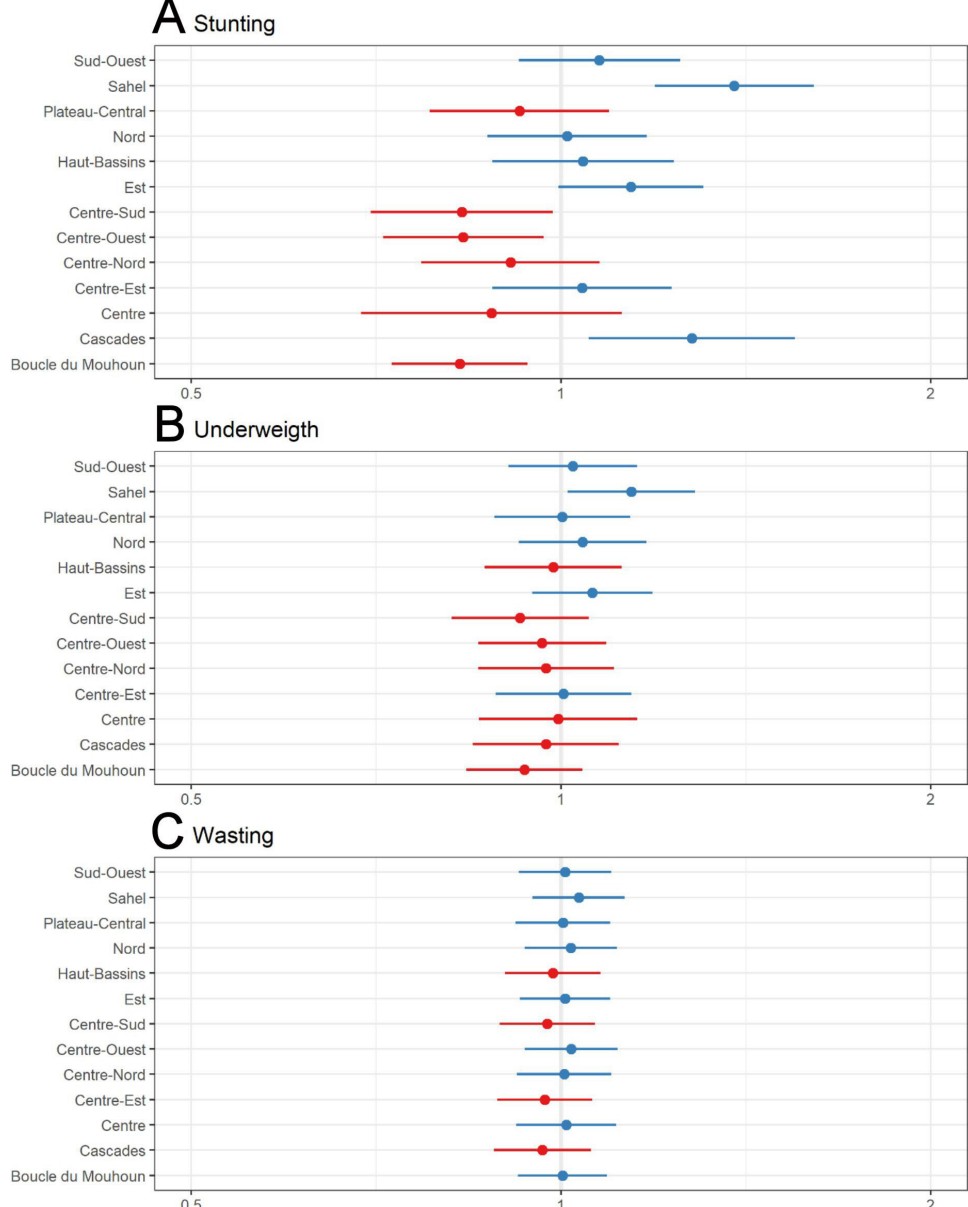

**Figure 2** Overall region-specific ORs (region random effect) of stunting (A), underweight (B) and wasting (C) among study population.

They are the result of a combination of several factors, including a short agricultural production season lasting 4 months of the year, climatic vulnerability, the depletion of natural resources, a chronic shortage of agricultural production and very little access to basic social services due to the weak state presence.

Insecurity marked by attacks from terrorist groups and the increase in inter-community conflicts exacerbate food and nutrition insecurity in the Sahel and North Central regions by reducing the already fragile livelihoods of populations that depend mainly on agriculture and pastoralism. Insecurity limits access to fields and agricultural production. The resulting road closures affect the supply of agricultural products to markets, reduce the availability of agricultural products, limit access due to price fluctuations, and hinder transhumance and access of animals to

pasture within Burkina and to Mali and Niger. The looting of food reserves and livestock reduces the resilience of communities affected by this insecurity. In addition, many health facilities are closed or operating at minimal capacity in areas affected by insecurity, reducing access to health services, including those for severely malnourished children. This has forced people to move to other parts of the country (almost 1 million internally displaced persons (IDPs) according to the United Nations Office for the Coordination of Humanitarian Affairs), resulting in overcrowding in host towns. While some IDPs reside in camps, more than 80% of IDPs are staying with host families whose food supplies are rapidly running out.

In our study, we found that poverty index is not associated with wasting, that mean acute undernutrition is due to an underlying disease causing abnormal nutrient loss,

increased energy expenditure or decreased food intake. Based on its aetiology, undernutrition is either illness related (one or more diseases or injuries directly result in nutrient imbalance) or caused by environmental/behavioural factors associated with decreased nutrient intake and/or delivery.[11 37 38] Therefore, primary acute undernutrition is mostly social rather than biomedical in origin, but it is also multifactorial. For example, poor water quality, sanitation and hygiene practices are increasingly believed to be the cause of the condition called 'environmental enteropathy' that contributes to acute undernutrition in childhood.[38]

Although this study provided insight on the temporal trends and overall spatial risk in undernutrition among children under-five between 2012 and 2018 in Burkina Faso, this study is influenced by several limitations. Our study did not cover all the NNSs from 2009 to 2021, due to data unavailability, therefore our finding is only relevant for the studied period. The analysis relies on survey data but not a cohort, hence causality cannot be inferred.

## Conclusion

The study found that undernutrition prevalence remains high and marked spatial disparities. Overall, there is a decrease in prevalence over the year. It is important to note that we found low-risk area where the control efforts need to be maintained and reinforced. The high-risk area, need to be allocated more resources and reinforce the control effort. Morbidity (fever and diarrhoea) constitutes important associated factors and need prompt and effective management with particular focus on children over 6 months. Further research should include birth cohort to allow causality inference for undernutrition in Burkina Faso. Additionally, surveillance system should be implemented towards IDPs. Our study showed that secondary analysis of national surveillance data has the potential to inform public health action and refinement of intervention.

**Author affiliations**
[1]Clinical Research Unit of Nanoro, Institut de Recherche en Sciences de la Sante, Nanoro, Burkina Faso
[2]Sydney Brenner Institute for Molecular Biosciences, University of the Witwatersrand Johannesburg, Johannesburg, South Africa
[3]Direction de la Nutrition, Ministère de la Santé, Ouagadougou, Burkina Faso
[4]Laboratoire de Biochimie, Biotechnologie, Technologie Alimentaire et Nutrition (LABIOTAN), Université Joseph Ki-Zerbo, Ouagadougou, Burkina Faso
[5]Institut National de Statistique et de la Démographie, Ouagadougou, Burkina Faso
[6]MRC Lifecourse Epidemiology Centre, University of Southampton, Southampton, UK
[7]Global Health Research Institute, University of Southampton, Southampton, UK

**Acknowledgements** The authors would like to thank the Ministry of Health and Public Hygiene for allowing access to the data.

**Collaborators** INPreP group: Abraham Oduro, Cornelius Debpuur, Doreen Ayibisah, Edith Dambayi, Engelbert Nonterah, Esmond W. Nonterah, James Adoctor, Josephine Addi, Maxwell Dalaba, Michael Banseh, Paul Welaga, Paula Beeri, Samuel Chatio, Winfred Ofosu (Navrongo Health Research Centre); Adélaïde Compaoré, Aminata Welgo, Kadija Ouedraogo, Karim Derra (Clinical Research Unit of Nanoro); Caroline Fall, Keith Godfrey, Mark Hanson, Mary Barker, Polly Hardy-Johnson, Sarah Kehoe (Medical Research Council Lifecourse Epidemiology Centre, University of Southampton); Marie-Louise Newell, Daniella Watson (Faculty of Medicine,

University of Southampton); Shane A Norris, Stephanie Wrottesley (SAMRC Developmental Pathways for Health Research Unit).

**Contributors** HS, M-LN, PB, HT and KW designed the WP5 study. HS, PB, HT and KW supervised the implementation of the study and guided the manuscript conception. PB and TR analysed the data, carried out the literature review and drafted the paper. EB, SK, EWRC, BO reviewed and approved the final manuscript. HS is the guarantor.

**Funding** This research was funded by the National Institute for Health Research (NIHR) (17/63/154) using UK aid from the UK Government to support global health research. The views expressed in this publication are those of the author(s) and not necessarily those of the NIHR or the UK Department of Health and Social Care.

**Map disclaimer** The inclusion of any map (including the depiction of any boundaries therein), or of any geographic or locational reference, does not imply the expression of any opinion whatsoever on the part of BMJ concerning the legal status of any country, territory, jurisdiction or area or of its authorities. Any such expression remains solely that of the relevant source and is not endorsed by BMJ. Maps are provided without any warranty of any kind, either express or implied.

**Competing interests** None declared.

**Patient and public involvement** Patients and/or the public were not involved in the design, or conduct, or reporting, or dissemination plans of this research.

**Patient consent for publication** Not applicable.

**Ethics approval** This study involves human participants and this study is part of the INPreP3 Study and has been approved by the National Health Ethics Committee in Burkina Faso, approval number 2018-12-156. Participants gave informed consent to participate in the study before taking part.

**Provenance and peer review** Not commissioned; externally peer reviewed.

**Data availability statement** Data are available upon reasonable request. Access to data is available upon request to the Ministry of Health and Public Hygiene.

**ORCID iD**
Palwende Romuald Boua http://orcid.org/0000-0001-8325-2665

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
