## [Reviewer comments · BMJ Open]

ARTICLE DETAILS

TITLE (PROVISIONAL)	Undernutrition in children aged 0-59 months by region and over time : Secondary analysis of the Burkina Faso 2012-2018 National Nutrition Surveys.
AUTHORS	Boua, Palwende; Rouamba, Toussaint; Bambara, Estelle; Kaboré, Saidou; Compaore, Ella W. R.; Ouedraogo, Boureima; Tinto, Halidou; Newell, Marie-Louise; Ward, Kate; Sorgho, Hermann

VERSION 1 – REVIEW

REVIEWER	Abayomi, Julie Edge Hill University, Department of Medicine and Nutrition
REVIEW RETURNED	17-Oct-2022

GENERAL COMMENTS	Thank you for asking me to review this interesting and well-written paper. I have only a few minor suggestions, that may help improve clarity. In general be consistent using 'undernutrition' rather than 'malnutrition'. It would be helpful to define and explain 'Food insecurity' in the introduction. If you have excluding children with oedema, you will be excluding children with kwashiorkor - so does this mean that your figures for undernutrition are underestimated? How many children were excluded for oedema? Some discussion of this would be helpful. I am not sure why you have excluded extreme values. Is this because they are not realistic and so you are presuming that they are an error? Please explain this. Is there any reason why undernutrition was more prevalent in males? Do you have an explanation for this? In the conclusion - I think you mean 'Over the years' (instead of 'throughout the year')? The conclusion needs to be stronger - I am not sure what you mean by 'sustain and reinforce' or by 'direct more resources and reinforce the control' - what does this mean? Can you be more specific. I suggest that the conclusion in the abstract could also be amended once the min conclusion is strengthened.
--

REVIEWER	Singh, Manisha University of Huddersfield, Department of Allied health professionals
REVIEW RETURNED	18-Nov-2022

GENERAL COMMENTS	Undernutrition is a very important topic. A very well put paper. However, please consider the following points for revision:
--

	1. Referecing: Please refer to the attached word file with detailed feedback on missing references. 2. Please proof read the manuscript once more paying attention to word economy. 3. Please strengthen your discussion section using feedback provided on the word file – see below. Section from manuscript Line(s) Comment/ suggested feedback Abstract 29 "Between 2012 and 2018, decreases were observed overall in the prevalence of stunting (stunting) decreased from 33.0% (95%CI: 32.3 – 33.8) in 2012to 26.7% (95%CI: 26.2 – 27.3) in 2018." Please rephrase this sentence. Unclear. Background While stunting and wasting prevalence data has been provided, please also provide incidence/prevalence data for underweight. 22 "In Burkina Faso, more than one-third of deaths among children under five years of age are directly or indirectly attributable to malnutrition. Despite the development and implementation by successive governments of numerous policies and programs to deal with malnutrition, the prevalence of undernutrition (stunting, underweight and wasting) remains high." Important point to mention. Please provide reference here. 32 "The government, supported by its partners, has..." Please clarify who do you mean by partners. 33 Please provide full form and brief description of "SMART methodology" on the first instance of use. Methods: Study setting 45-47 Please use uniform in-text referencing style. In the rest of the manuscript, numbers have been used to indicate references whereas in lines 45-47, author and year have been mentioned. 51-52 Please provide reference. 55 "It is estimated that approximately 3,280,800....." Please report similar statistics uniformly. Use 3.28 million instead of total figure. Methods: Data sources 16-17 For this study, we focus on anthropometric measurements collected between 2012 and 2018 (seven surveys) among population aged 0-59 months." Would you please clarify/ justify why you chose six years for the analysis? Why 2012-2018? 17 "....among population aged 0-59 months" Replace the word population with children. Also, please use either children under 5 or 0-59 months consistently. 50-59 Reconsider this paragraph and explain the implications of different level of representativeness for generalizability.
--	--

	Results: Demographic characteristics and anthropometrical measures of children 3-4 "Gender was equally distributed through the surveys and varied between 51.6% and 51.1% for the proportion of male children. The average age of the children during the period ranged from 27 to 29 months" Please provide the years for this data. Results: Potential risk factors associated with Stunting, Underweight, and Wasting from 2012 to 2017 in Burkina Faso 9 "In multivariable analysis, allowing for xxxxxx (Table 3)..." Please insert missing figures. Discussion 39-40 "the spatial risk to experience an undernutrition event is not uniform throughout the country" Rephrase for clarity 44-46 Please explain possible reasons for discrepancy between your findings and that predicted by literature. 57 "This means that access to health care is associated with nutritional status." Without statistical analysis, you cannot use the term "association" to describe relationship between variables. I suggest either rephrasing to "this suggests that health care could be associated with nutritional status" OR provide reference of a study/studies that have confirmed this association statistically. 58-60 "The persistent prevalence of undernutrition translates a need, not just for the community management of wasting programmes which have been shown to be effective, but for effective prevention programmes 31" I suggest finding more evidence to support this point and recent publications. Some recent articles: https://equityhealthj.biomedcentral.com/articles/10.1186/s12939-020-01258-5 https://bmjopen.bmj.com/content/9/9/e029968 https://www.sciencedirect.com/science/article/pii/S2667268522000092 Pg 10, Discussion Line 3-30 Please provide references in these two paragraphs. The issues such as food security have already been identified as risk factors of malnutrition, using those references here will only strengthen your discussion/findings. Also, it would be better if you would defend your key findings by comparing with current literature. In this paragraph you talk about reasons for food insecurity. Is food insecurity your key finding though? Please start each discussion paragraph/section with your key finding and then defend/compare with literature. For example, first mention/repeat your undernutrition data/findings on Sahel and Northern region and then elaborate on food insecurity and terrorism as explanations for your findings in these regions. 44-45
--	---

	“Our study did not cover all the National Nutrition Surveys from 2009 to 2021, due to data availability...” Do you mean data unavailability? “It is important to note that we found low risk area that need to sustain and reinforce the control effort. The high-risk area, need to be directed more resources and reinforce the control effort. “ Please rephrase the sentences.
--	---

REVIEWER	Bhusal, Umesh Prasad Public health and social protection professional
REVIEW RETURNED	17-Jan-2023

GENERAL COMMENTS	I congratulate authors for carrying out this important research on policy relevant topic. In general, I found this research was carried out very carefully. However, there are few issues outlined below that must be addressed before it could be considered a statistically rigorous study.  1. Authors must provide broader framework or theory that derived the choice of independent variables. Authors have modelled occurrence of diarrhoea and fever among children two weeks prior to date of data collection. However, it looks very inconvenient that these variables could explain the outcomes of chronic undernutrition such as stunting in children. Please explain why such variables were used. Authors have missed others potential variables such as maternal health characteristics and health services usage by mothers in their models. So, the authors should base their regression model based on child nutrition framework and conduct rigorous literature review to come up with set of independent variables. 2. Please explain how poverty index was measured. 3. Please provide justification for using GLMM. At the moment the authors have not reported the fitness of the regression model they have used. It is a must that the regression diagnostics be conducted, and the results should be provided as supplementary materials so that the model specification related concerns are properly addressed. 4. Readers may want to know why you did not account for survey design while conducting regression analyses. 5. Please provide what assumptions of GLMM that you needed to test before implementing such analyses and provide the summary as supplementary materials. 6. The sample size of 2016 looks substantially lower than other survey years. It would be helpful if authors could provide some note below the Table 2. 7. Please mention in the Method section how the surveys from different years were linked (appended?). Did you have to adjust for factors that were different across different years? Please also consider substantially low sample size of year 2016. 8. Minor issues: please check line 9 of page 8. Line 12 of the same page need some grammatical correction. Similarly, please check why the reference for the ‘age in month’ of table 3 is missing. Line 46, page 9 please mention unit as well after 6.7. 9. Table 3 shows fever and diarrhoea in children two weeks prior to survey were statistically significant with the outcome of chronic undernutrition. Please recheck, such findings can be due to possible bias in selection of explanatory variables for your regression model. 10. In the discussion section please consider discussing about the
--

	strength and limitation of your study design in terms of methodological choices that you have made. 11. Please recheck whether the last line of the conclusion is related to your research objective and the findings. 12. Figure 5 and 6: You have mentioned relative risk but, in the result, you have provided ORs. Please recheck or clarify
--	---

VERSION 1 – AUTHOR RESPONSE

Reviewer: 1

Dr. Julie Abayomi, Edge Hill University

Comments to the Author:

Thank you for asking me to review this interesting and well-written paper. I have only a few minor suggestions, that may help improve clarity.

In general be consistent using 'undernutrition' rather than 'malnutrition'.

We did replace all malnutrition by undernutrition.

It would be helpful to define and explain 'Food insecurity' in the introduction.

We add a phrase defining “food insecurity” at the end of the second paragraph of the introduction.

If you have excluding children with oedema, you will be excluding children with kwashiorkor - so does this mean that your figures for undernutrition are underestimated? How many children were excluded for oedema? Some discussion of this would be helpful.

The exclusion of the children with oedema does not necessarily lead to an underestimation. They are excluded because it is not possible to calculate Weight-for-Height and Weight-for-Age z-scores for them. The number of children excluded for oedema is 2138 (1.6% of total sample). This is in line with the criteria recommended for analysis of nutrition surveys data.

I am not sure why you have excluded extreme values. Is this because they are not realistic and so you are presuming that they are an error? Please explain this.

In each of the strata surveyed, WHO flags were used to exclude data; It is the z-score values of Weight-for-Height <-5 or >+5, Height-for-- Age <-6 or >+6, Weight-for-Age <-6 or >+5 (WHO, 2006) that were excluded. This is in line with the criteria recommended for analysis of nutrition surveys data.

Is there any reason why undernutrition was more prevalent in males? Do you have an explanation for this?

We added a sentence to the discussion: “Biological and social mechanism or both has been proposed to explain the differences between boys and girls (Thurstans et al., 2020)”

In the conclusion - I think you mean 'Over the years' (instead of 'throughout the year')?

Thank you for the correction.

The conclusion needs to be stronger - I am not sure what you mean by 'sustain and reinforce' or by 'direct more resources and reinforce the control' - what does this mean? Can you be more specific. I suggest that the conclusion in the abstract could also be amended once the main conclusion is strengthened.

We revised these sentences: "It is important to note that we found low risk area where the control effort need to maintained and reinforced. The high-risk area, need to be allocated more resources and reinforce the control effort."

We added a sentence to the conclusion: "Our study showed that secondary analysis of national surveillance data have the potential to inform public health action and refinement of intervention."

Reviewer: 2

Miss Manisha Singh, University of Huddersfield

Comments to the Author:

Undernutrition is a very important topic. A very well put paper. However, please consider the following points for revision:

1. Referencing: Please refer to the attached word file with detailed feedback on missing references.
2. Please proof read the manuscript once more paying attention to word economy.
3. Please strengthen your discussion section using feedback provided on the word file.

"Between 2012 and 2018, decreases were observed overall in the prevalence of stunting (stunting) decreased from 33.0% (95%CI: 32.3 – 33.8) in 2012to 26.7% (95%CI: 26.2 – 27.3) in 2018." Please rephrase this sentence. Unclear

We have rephrased this sentence for more clarity: "Between 2012 and 2018, decreases were observed overall in the prevalence of growth retardation (stunting) decreased..."

"The government, supported by its partners, has..." Please clarify who do you mean by partners.

We rephrased the sentence: "The government, supported by its financial and technical partners"

Please provide full form and brief description of "SMART methodology" on the first instance of use.

We provided the full form of SMART on first instance of use and a brief description : "...SMART (Standardized Monitoring and Assessment of Relief and Transitions) methodology. This methodology is widely used by governments and humanitarian partners to conduct timely nutrition surveys in all contexts (emergency, development, displaced populations). SMART surveys are conducted on a regular basis, often in connection with seasonal malnutrition, and can be conducted at the national or regional level, and even on a smaller scale. "

Please use uniform in-text referencing style. In the rest of the manuscript, numbers have been used to indicate references whereas in lines 45-47, author and year have been mentioned.

Thank you to the reviewer for their attention, this has been corrected (Ref. 21).

"It is estimated that approximately 3,280,800....." Please report similar statistics uniformly. Use 3.28 million instead of total figure.

Corrected as suggested

For this study, we focus on anthropometric measurements collected between 2012 and 2018 (seven surveys) among population aged 0-59 months." Would you please clarify/ justify why you chose six years for the analysis? Why 2012-2018?

The focus of our analysis on the period between 2012 and 2018 was guided by the availability of the datasets.

"....among population aged 0-59 months" Replace the word population with children. Also, please use either children under 5 or 0-59 months consistently.

Corrected as suggested

Reconsider this paragraph and explain the implications of different level of representativeness for generalizability.

We thank the reviewer for their recommendation, we have provided explanation of implications and how the different level of representativeness guided the choice of method in the statistical analysis section.

"Gender was equally distributed through the surveys and varied between 51.6% and 51.1% for the proportion of male children. The average age of the children during the period ranged from 27 to 29 months". Please provide the years for this data

We have revised the phrase to : "The average age of the children during the period of 2012 to 2018 ranged from 27 to 29 months"

"In multivariable analysis, allowing for xxxxxx (Table 3)..." Please insert missing figures.

Corrected to : "In multivariable analysis, allowing for adjustment (Table 3)..."

"the spatial risk to experience an undernutrition event is not uniform throughout the country" Rephrase for clarity

Corrected to : "The analysis found that undernutrition remains very high in certain part of the country, although there was an overall decrease in prevalence and risk over the years in the country; the geospatial risk to experience an undernutrition event is not uniform throughout the country."

Please explain possible reasons for discrepancy between your findings and that predicted by literature

We added to the discussion: "De Onis prediction were based on poor data from some countries with lag time of about ten years between them (namely Demographic and Health Survey), and the landscape of interventions in nutrition was poor. From 2009, a nutrition surveillance system has been implemented with yearly national surveys, additionally more effective nutrition programs to improve nutritional status across the whole population have been implemented."

"This means that access to health care is associated with nutritional status.". Without statistical analysis, you cannot use the term "association" to describe relationship between variables. I suggest either rephrasing to "this suggests that health care could be associated with nutritional status" OR provide reference of a study/studies that have confirmed this association statistically.

Corrected as suggested

“The persistent prevalence of undernutrition translates a need, not just for the community management of wasting programmes which have been shown to be effective, but for effective prevention programmes 31” I suggest finding more evidence to support this point and recent publications.

We have amended the phrase and added references for support of statement: “The persistent prevalence of undernutrition translate a need, not just for the community management of wasting programs which have been shown to be effective, but for effective prevention programs as well as structural changes allowing reduction of inequalities”

“Our study did not cover all the National Nutrition Surveys from 2009 to 2021, due to data availability...” Do you mean data unavailability?

Corrected to “unavailability”

“It is important to note that we found low risk area that need to sustain and reinforce the control effort. The high-risk area, need to be directed more resources and reinforce the control effort. “ Please rephrase the sentences.

We revised these sentences: “It is important to note that we found low risk area where the control effort need to maintained and reinforced. The high-risk area, need to be allocated more resources and reinforce the control effort.”

Reviewer: 3

Mr. Umesh Prasad Bhusal, Public health and social protection professional

Comments to the Author:

I congratulate authors for carrying out this important research on policy relevant topic. In general, I found this research was carried out very carefully. However, there are few issues outlined below that must be addressed before it could be considered a statistically rigorous study.

1. Authors must provide broader framework or theory that derived the choice of independent variables. Authors have modelled occurrence of diarrhoea and fever among children two weeks prior to date of data collection. However, it looks very inconvenient that these variables could explain the outcomes of chronic undernutrition such as stunting in children. Please explain why such variables were used. Authors have missed others potential variables such as maternal health characteristics and health services usage by mothers in their models. So, the authors should base their regression model based on child nutrition framework and conduct rigorous literature review to come up with set of independent variables.

To be responded

2. Please explain how poverty index was measured.

We did not generate the index, we did use the poverty index as published by Zida and Kambou, 2014.

3. Please provide justification for using GLMM. At the moment the authors have not reported the fitness of the regression model they have used. It is a must that the regression diagnostics be conducted, and the results should be provided as supplementary materials so that the model

specification related concerns are properly addressed.

GLMM are used to analyze non-independent, grouped, or hierarchical data. In our case GLMM are useful to handle survey data because they can include several sources affecting the responses in a model with fixed (observed individual characteristics) or random effects, and the distribution of responses is not limited to a normal distribution. In this study, we considered that the data have a hierarchical structure, and we would need to account for the cluster (province and region) variability as well as the individual-to-individual variability.

4. Readers may want to know why you did not account for survey design while conducting regression analyses.

Many thanks for the suggestion. This is taken into-account in the current updated version of the manuscript.

The sentence is formulate as follow "Since we intended to quantify the association between the outcomes and dependent variables by avoiding that the coefficient be affected by the individual weight, survey weights were not used to adjust the regressions and a complete case analysis was used"

5. Please provide what assumptions of GLMM that you needed to test before implementing such analyses and provide the summary as supplementary materials.

We considered that the data have a hierarchical structure, and we would need to account for the cluster (province and region) variability as well as the individual-to-individual variability. We assume the response was drawn from a binomial distribution the residual are independent and normally distributed. The random effects are also, independent and normally distributed. The model diagnostics plots were presented in supplementary materials.

6. The sample size of 2016 looks substantially lower than other survey years. It would be helpful if authors could provide some note below the Table 2.

We added a note below Table 2: "In 2016 the sampling framework was regional and national (without provincial), hence impacting the sample size. This choice by the Ministry of Health was guided by limited resources to run the national survey."

7. Please mention in the Method section how the surveys from different years were linked (appended?). Did you have to adjust for factors that were different across different years? Please also consider substantially low sample size of year 2016.

The different datasets were appended to build the final database.

To adjust for factors that were different across different years, we included in our model the year as covariate (fixed model) and also, consider the year as random effect.

For substantially low sample size of year 2016, we have taken into account the study design and the survey weight when it comes to estimates the national estimates.

For the regression, since we intended to quantify the association between the outcomes and dependent variables by avoiding that the coefficient be affected by the individual weight, survey weights were not used to adjust the regressions and a complete case analysis was used.

8. Minor issues: please check line 9 of page 8. Line 12 of the same page need some grammatical correction. Similarly, please check why the reference for the 'age in month' of table 3 is missing. Line

46, page 9 please mention unit as well after 6.7.

Corrected as suggested on Table 3

9. Table 3 shows fever and diarrhoea in children two weeks prior to survey were statistically significant with the outcome of chronic undernutrition. Please recheck, such findings can be due to possible bias in selection of explanatory variables for your regression model.

We did not intend to test the association between fever and diarrhoea and chronic undernutrition. Those are often associated with acute undernutrition which we found in our study. Nonetheless, it appeared associated and this seen as the fact that children who will suffer from multiple infections or health threatening conditions will be more likely to suffer from chronic undernutrition.

10. In the discussion section please consider discussing about the strength and limitation of your study design in terms of methodological choices that you have made.

The limitations have been discussed in the last paragraph of the discussion.

11. Please recheck whether the last line of the conclusion is related to your research objective and the findings.

We added a sentence to the conclusion: "Our study showed that secondary analysis of national surveillance data have the potential to inform public health action and refinement of intervention."

12. Figure 5 and 6: You have mentioned relative risk but, in the result, you have provided ORs. Please recheck or clarify

We have checked and it was a mistake, we intend to refer to Odd ratio. We thank the reviewer for their attention.

VERSION 2 – REVIEW

REVIEWER	Bhusal, Umesh Prasad Public health and social protection professional
REVIEW RETURNED	15-May-2023

GENERAL COMMENTS	Many thanks for submitting the revised version of the manuscript. You have addressed many of the comments provided by reviewers in the earlier revision. I found you that not yet responded to my comment number 1. To avoid delay in the decision making process I suggest authors following: 1. Please go through each and every comment of all the reviewers once more to ensure you have provided sufficient response and made necessary changes in the manuscript. 2. Please review the manuscript to ensure it is free from typos and grammatical errors 3. Please review all the tables and figures (both from the manuscript and supplementary resources) and make sure they are well presented and adequately described. I wish you all the best!
---

VERSION 2 – AUTHOR RESPONSE

Reviewer: 3

Mr. Umesh Prasad Bhusal, Public health and social protection professional

Comments to the Author:

Dear authors,

Many thanks for submitting the revised version of the manuscript. You have addressed many of the comments provided by reviewers in the earlier revision.

I found you that not yet responded to my comment number 1.

To avoid delay in the decision making process I suggest authors following:

1. Please go through each and every comment of all the reviewers once more to ensure you have provided sufficient response and made necessary changes in the manuscript.

Thank you to the reviewer, we provide below the response to their first comment in review 1

Comment

Authors must provide broader framework or theory that derived the choice of independent variables. Authors have modelled occurrence of diarrhoea and fever among children two weeks prior to date of data collection. However, it looks very inconvenient that these variables could explain the outcomes of chronic undernutrition such as stunting in children. Please explain why such variables were used. Authors have missed others potential variables such as maternal health characteristics and health services usage by mothers in their models. So, the authors should base their regression model based on child nutrition framework and conduct rigorous literature review to come up with set of independent variables.

Response

This paper is a secondary data analysis of the National Nutrition Surveys, therefore our analysis is limited to the availability of variables collected during this surveys, therefore we could not include maternal health and health service usage by mothers. Diarrhoea and fever were indicators of child morbidity rather than causes of undernutrition.

2. Please review the manuscript to ensure it is free from typos and grammatical errors

We have reviewed the manuscript for typos and grammatical errors

3. Please review all the tables and figures (both from the manuscript and supplementary resources) and make sure they are well presented and adequately described.

We have reviewed the manuscript the tables and figures